# Param$\Delta$ for Direct Weight Mixing: Post-Train Large Language Model at Zero Cost

**Sheng Cao**
Meta Platforms, Inc
rickcao@meta.com

**Mingrui Wu**
Meta Platforms, Inc
mingruiwu@meta.com

**Karthik Prasad**
Meta Platforms, Inc
krp@meta.com

**Yuandong Tian** [*]
Meta FAIR
yuandong@meta.com

**Zechun Liu** [*]
Meta Reality Labs
zechunliu@meta.com

## Abstract

The post-training phase of large language models is essential for enhancing capabilities such as instruction-following, reasoning, and alignment with human preferences. However, it demands extensive high-quality data and poses risks like overfitting, alongside significant computational costs due to repeated post-training and evaluation after each base model update. This paper introduces Param$\Delta$, a novel method that streamlines post-training by transferring knowledge from an existing post-trained model to a newly updated base model with **zero** additional training. By computing the difference between post-trained model weights ($\Theta_{\text{post}}$) and base model weights ($\Theta_{\text{base}}$), and adding this to the updated base model ($\Theta'_{\text{base}}$), we define Param$\Delta$ Model as: $\Theta_{\text{Param}\Delta} = \Theta_{\text{post}} - \Theta_{\text{base}} + \Theta'_{\text{base}}$. This approach surprisingly equips the new base model with post-trained capabilities, achieving performance comparable to direct post-training. We did analysis on LLama3, Llama3.1, Qwen, and DeepSeek-distilled models. Results indicate Param$\Delta$ Model effectively replicates traditional post-training. For example, the Param$\Delta$ Model obtained from 70B Llama3-inst, Llama3-base, Llama3.1-base models attains approximately 95% of Llama3.1-inst model's performance on average. Param$\Delta$ brings a new perspective on how to fully leverage models in the open-weight community, where checkpoints for base and instruct models are readily available and frequently updated, by providing a cost-free framework to accelerate the iterative cycle of model development.

## 1 Introduction

The advent of Large Language Models (LLMs) has ushered in a new era for Natural Language Processing (NLP), redefining the boundaries of what machines can achieve in understanding and generating human languages. Frontier models such as GPT-4 (Achiam et al., 2023) Claude (Anthropic, 2023), and Gemini (Team et al., 2023) have not only set new benchmarks in performance but also led to a pivotal shift in AI research and application. Recently, reasoning models like OpenAI-o1 (OpenAI, 2024) and Deepseek-R1 (Guo et al., 2025) have refreshed the limit of models with advanced post-training techniques.

The unprecedented success of LLMs stems from their sophisticated training process, which includes pretraining and post-training (usually refers to instruction-finetuning and reasoning-enhanced reinforcement learning), where post-training can be further categorized into general-purpose post-training and task-specific post-training. Although the emergence of open-weight models, such as Llama (Dubey et al., 2024), Mistral (Jiang et al., 2023), Gemma (Team et al., 2024), DeepSeek (Guo et al., 2025; Liu et al., 2024) and Qwen (Yang et al., 2024), has alleviated some of the burdens associated with pretraining and general-purpose post-training, the task-specific post-training remains a resource-intensive and complex process. Furthermore, with the rapid release of updated

---

[*]Equal Advising

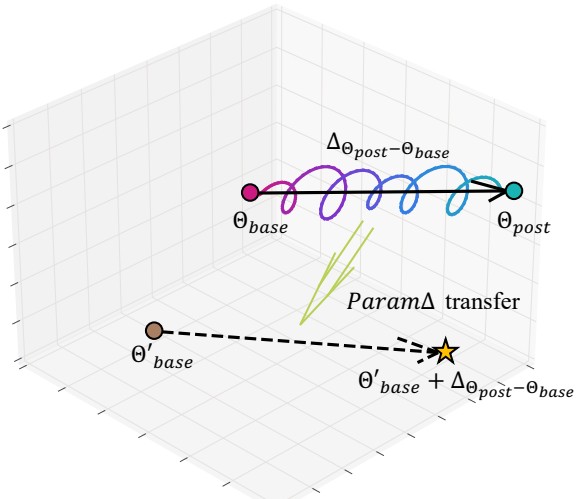

Figure 1: Overall diagram of Param$\Delta$. Given a newly updated base model $\Theta'_{\text{base}}$, we can obtain its post-trained version with no additional training cost, by simply adding the weight difference between the previous post-trained model $\Theta_{\text{post}}$ and original base model $\Theta_{\text{base}}$: $\Theta'_{\text{post}} = \Theta'_{\text{base}} + (\Theta_{\text{post}} - \Theta_{\text{base}})$.

base models by both foundational companies and the open-source community, almost monthly, previously post-trained models on older versions quickly become outdated and lose their state-of-the-art status. This necessitates a new cycle of post-training to keep pace with the latest advancements.

Unlike the pretraining phase, which typically uses unlabeled general-purpose data, the post-training phase poses greater challenges in two key areas: 1) **Data Requirements**: The need for large volumes of high-quality, instruction-aligned supervised training data can be time-consuming and costly, creating a significant barrier to effective post-training. Reinforcement learning even requires a reliable reward system (Ouyang et al., 2022) trained on labeled data; 2) **Sophisticated Post-training Techniques**: Instruction finetuning without overfitting and losing generalization capabilities is challenging. Additionally, the advent of reasoning-enhanced reinforcement learning has further complicated post-training methodologies.

In this paper, we present Param$\Delta$, a *simple yet surprisingly effective* approach that transfers knowledge from an existing post-trained model to a newly upgraded base model **without additional training**. Our method starts with calculating the parameter difference between the base pre-trained and post-trained checkpoints ($\Delta\Theta = \Theta_{\text{post}} - \Theta_{\text{base}}$). This difference is hypothesized to "store" the data-specific knowledge and capabilities acquired during post-training. We then apply $\Delta\Theta$ to an updated checkpoint of base model with same architecture ($\Theta'_{\text{base}}$) to yield a new post-trained checkpoint ($\Theta'_{\text{post}} = \Theta'_{\text{base}} + \Delta\Theta$). Surprisingly, such a simple approach yields a strong Param$\Delta$ Model, which achieves comparable to a traditional post-trained model in various tasks and different scenarios. For example, the Param$\Delta$ Model obtained from 70B Llama3-inst, Llama3-base, Llama3.1-base models attains approximately 95% of Llama3.1-inst model's performance on average. This work draws inspiration from studies on model weights averaging (Izmailov et al., 2018; Nikishin et al., 2018; Rame et al., 2022; Su & Chen, 2015; Wortsman et al., 2022a) and model merging (Ilharco et al., 2022; Yadav et al., 2024; Yu et al., 2024).

Our contributions are as follows: 1) We propose a training-free solution that bypasses the traditional computation-heavy and data-intensive post-training process; 2) We identify four representative scenarios (Figure 4) of model development cycles in industry settings and provide best-practice guidelines for applying Param$\Delta$ approach to each scenario; 3) We perform a comprehensive evaluation of the Param$\Delta$ method across the Llama, Qwen and DeepSeek-distilled models, demonstrating its transfer effectiveness, robustness, and ease of integration for real-world applications.

## 2 METHODOLOGY

### 2.1 PARAMETER SPACE AND PARAMETER DELTA

**Parameter Space** refers to a high-dimensional space composed of the sets of parameters associated with homologous models. The homologous models encompass the same model configuration, such

as the number of layers, number of heads, and number of dimensions, as well as the tokenizer. Parameter space and text embedding space (Aggarwal & Zhai, 2012; Angelov, 2020; Church, 2017) can be considered analogous. Just as the embedding space conveys the meanings and relationships inherent in text-based data, the model parameter space embodies the model's comprehension of the tasks or the knowledge it has accrued through training.

**Parameter Delta** is the subtraction of model parameters between two checkpoints within homologous models and is expressed as $\Delta\Theta$.

**Parameter Fusing** refers to the process of mixing parameters or parameter deltas. These operations typically involve adding and subtracting the model parameters.

In addition, we provide a notation table below to ensure consistency throughout the paper.

| Notation | Description |
|---|---|
| $\Theta$ | Model parameters |
| $f(\Theta)$ | Model outputs |
| $\Theta_0, \Theta_{\text{base}}$ | Pretrained or base model parameters |
| $\Theta_{\text{cpt}}$ | Continual pretrained model parameters |
| $\Theta_{\text{post}}$ | Post-trained or instruction-finetuned model parameters |
| $\Delta\Theta$ | Parameter delta between a post-trained model and a base model |
| $\mathbb{T}$ | Amount of training received |
| $\alpha, \beta$ | Scaling factor of parameter delta $\Delta\Theta$ |
| $\gamma$ | Coefficient of knowledge transfer efficiency |
| Param$\Delta$ Model | The final model obtained by adding $\Delta\Theta$ |

## 2.2 KNOWLEDGE TRANSFER THROUGH MIXING EXISTING MODELS' PARAMETERS

We hypothesize that the language model can acquire the knowledge and capabilities characteristic of the post-training stage by incorporating the parameter differences between a post-trained checkpoint and a base checkpoint. Conventionally, the foundational knowledge of a large language model is established through initial pretraining from scratch. This is followed by a post-training process, which includes supervised finetuning (SFT) (Brown, 2020; Dubey et al., 2024; Touvron et al., 2023) and reinforcement learning including DPO (Dubey et al., 2024; Rafailov et al., 2024; Touvron et al., 2023), PPO (Ouyang et al., 2022; Schulman et al., 2017) or GRPO (Shao et al., 2024).

Our proposed approach bypasses the standard post-training process and instead integrates the available post-trained model's parameter deltas into the new base model, as the parameter deltas encapsulate the embedded knowledge derived from the underlying training data and reflect the cumulative gradient changes resulting from the volume of training tasks. This can be represented as

$$\Theta_{\text{Param}\Delta} = \Theta'_{\text{base}} + \Delta\Theta \longleftrightarrow \mathbb{T}_{\text{pretrain}} + \mathbb{T}_{\text{post-train}} \tag{1}$$

where $\Theta'_{\text{base}}$ is an updated base model, either from newly pretrained or continually pretrained. $\Delta\Theta$ is calculated from $\Delta\Theta = \Theta_{\text{post}} - \Theta_{\text{base}}$ with pre-existing model checkpoints, $\mathbb{T}_{\text{post-train}}$ symbolizes the post-training process.

## 2.3 AN EMPIRICAL ANALYSIS ON THE HYPOTHESIS AND EFFECTIVENESS OF PARAM$\Delta$

To elucidate the structure of information encoding within the weight space 2.2, we can infer and then hypothesize that distinct clusters of information are embedded in mutually orthogonal subspaces of the parameter space. Furthermore, we posit that the relevance of weight perturbations exhibits a positive correlation with the relevance of the corresponding training data, suggesting an intrinsic alignment between weight-space dynamics and the encoded training data semantics. To verify this hypothesis, we visualize the weight delta orthogonality and the weight norms from different layers (attention layers and feed-forward layers) of various post-trained models including Llama, Qwen and DeepSeek-distilled series. We discovered several interesting findings: 1) the parameter differences ($\Delta\Theta$) tend to be orthogonal (i.e., exhibit close-to-zero cosine similarities), when there is minimal overlap in the datasets used during post-training. For example, we can see from Figure 2 (a), the cosine similarity between the parameter difference of the DeepSeek-R1 distilled 8B model $\Delta\Theta = (\Theta_{\text{DeepSeek-R1-Distill-Llama-8B}} - \Theta_{\text{Llama3.1-8B-base}})$ and the parameter difference of the

Llama3.1-8B instruction-finetuned model $\Delta\Theta = (\Theta_{\text{Llama3.1-8B-inst}} - \Theta_{\text{Llama3.1-8B-base}})$ is close to zero. Similar trends can be found in each pair between parameter differences of DeepSeek-R1 distilled Llama3 models and Llama3-inst, Llama3.1-inst models, or a medical-specific model Bio-Medical-Llama-3-8B (Con, 2024) [1]; 2) There is a notable similarity between the parameter differences of the Llama3-inst and Llama3.1-inst models, attributable to the shared datasets employed during their post-training processes, which can be observed in Figure 2 (c) (g); 3) The norms of the parameter differences in the feed-forward layers consistently exceed those in the attention layers (Figure 3). This indicates that a significant portion of the learned knowledge is embedded within the feed-forward layers, aligning with findings from previous research (Geva et al., 2020). We extended our analysis to the Qwen-series models (Yang et al., 2024), as illustrated in Figures 7 and 8, and observed similar patterns and conclusions.

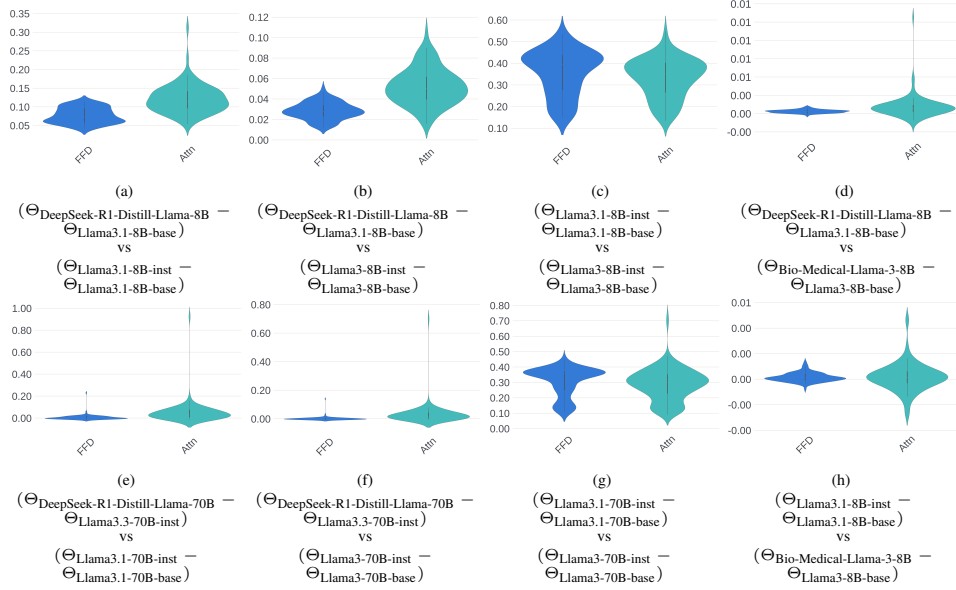

Figure 2: Cosine similarities of parameter differences from the feed-forward layers and attention layers from various post-trained Llama-series models and their corresponding base models.

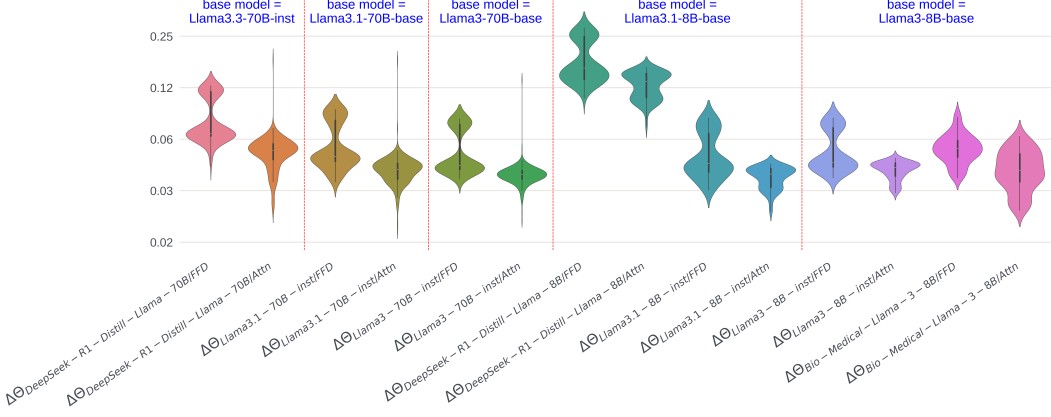

Figure 3: Weight norms distribution of parameter differences from the feed-forward layers and attention layers from various post-trained Llama-series models and their corresponding base models.

## 3 RECIPES ON PARAM$\Delta$

In particular, we highlight four representative scenarios and their corresponding recipes in Figure 4 that can benefit from our approach. All scenarios anchors on the pretrained base checkpoint while

---

[1]The base model for DeepSeek-R1-Distill-Llama-70B is Llama3.3-70B-inst and the base model for DeepSeek-R1-Distill-Llama-8B is Llama3.1-8B-base.

adding the $\Delta\Theta$ from different post-trained checkpoints. Readers may identify optimal use cases for applying Param$\Delta$.

*Scenario 1* **General-purpose post-training**: $\Theta'_{\text{base}} + \Delta\Theta^{\text{general}}_{\text{post}}$, where $\Delta\Theta^{\text{general}}_{\text{post}} = \Theta^{\text{general}}_{\text{post}} - \Theta_{\text{base}}$ When a newly updated base model is available, existing $\Delta\Theta$ from the last instruction-finetuned models can be integrated to bypass general-purpose post-training on the new base model. This approach is particularly advantageous for foundational model companies that produce both base models and instruction-finetuned models, as it reduces the costs of repetitive general-purpose post-training.

*Scenario 2* **Task-specific post-training**: $\Theta'_{\text{base}} + \Delta\Theta^{\text{specific}}_{\text{post}}$. In the event of a newly updated base model, task-specific post-training can be circumvented by incorporating $\Delta\Theta$ from existing task-specific-tuned models. Scenario 2 differs from Scenario 1 in the nature of $\Delta\Theta$: Scenario 1 utilizes a general-purpose post-trained checkpoint, while Scenario 2 employs a task-specific post-trained checkpoint. Scenario 2 is especially beneficial for individuals or application companies focusing on specialized use cases, enabling them to swiftly update their previously task-specific-tuned models to align with the new foundational base model.

*Scenario 3* **Continual pretraining**: $\Theta^{\text{continue}}_{\text{base}} + \Delta\Theta_{\text{post}}$. Continual pretraining followed by instruction finetuning is advocated as an effective strategy for domain adaptation by several researchers (Cossu et al., 2024; Guo et al., 2024; Ke et al., 2023). Historically, this approach has been met with apprehension and concern due to the substantial data and GPU requirements associated with the subsequent post-training process. However, in numerous situations where constraints such as the **lack of labeled training data** are present, self-supervised learning may be the only viable means for a model to acquire knowledge. In this context, our proposed approach offers a solution to this challenge: after learning knowledge during the continual pretraining phase, the need for post-training can be bypassed by integrating $\Delta\Theta$ from existing models. This approach is especially advantageous for the development of domain-specific foundational models, as it enables the model to initially acquire knowledge and subsequently swiftly obtain all necessary capabilities from a general-purpose or task-specific post-training process, thereby obviating the need for repetitive instruction-finetuning or reinforcement learning.

*Scenario 4* **Combining knowledge from multiple post-training sources** such as general-purpose post-training and task-specific post-training: $\Theta'_{\text{base}} + \alpha\Delta\Theta^{\text{general}}_{\text{post}} + \beta\Delta\Theta^{\text{specific}}_{\text{post}}$. Scenario 4 serves as an extended application of Scenario 2. When both a newly updated base model and its corresponding post-trained model are available, $\Delta\Theta$ can be employed not only to circumvent task-specific post-training but also to integrate the enhanced capabilities of the new post-trained models. Analogous to Scenario 2, this methodology provides significant benefits to individuals or organizations that regularly update their domain-specific foundational models.

Our approach not only improves the efficiency of the development process but also has the potential to promote greater adoption of open-weight models, especially those subject to frequent updates. By streamlining the integration of new advancements, Param$\Delta$ can enhance accessibility and encourage the widespread utilization of state-of-the-art models within the research and development community.

## 4 EXPERIMENTS

### 4.1 EXPERIMENTS SETUP

We utilize open-weight checkpoints of Llama3 and Llama3.1 (Dubey et al., 2024). The model parameter delta for post-training is defined as $\Delta\Theta = \Theta_{\text{post}} - \Theta_{\text{base}}$. The evaluation datasets employed are sourced from open-source collections as reported in the Llama3 paper, which include MMLU(Hendrycks et al., 2021a), IFEval(Zhou et al., 2023), HumanEval(Chen et al., 2021), MBPP(Austin et al., 2021), GSM8K(Cobbe et al., 2021), MATH(Hendrycks et al., 2021b), ARC Challenge (Clark et al., 2018), GPQA (Rein et al., 2023), BFCL(Yan et al., 2024), API-Bank(Li et al., 2023), and MGSM(Shi et al., 2022).

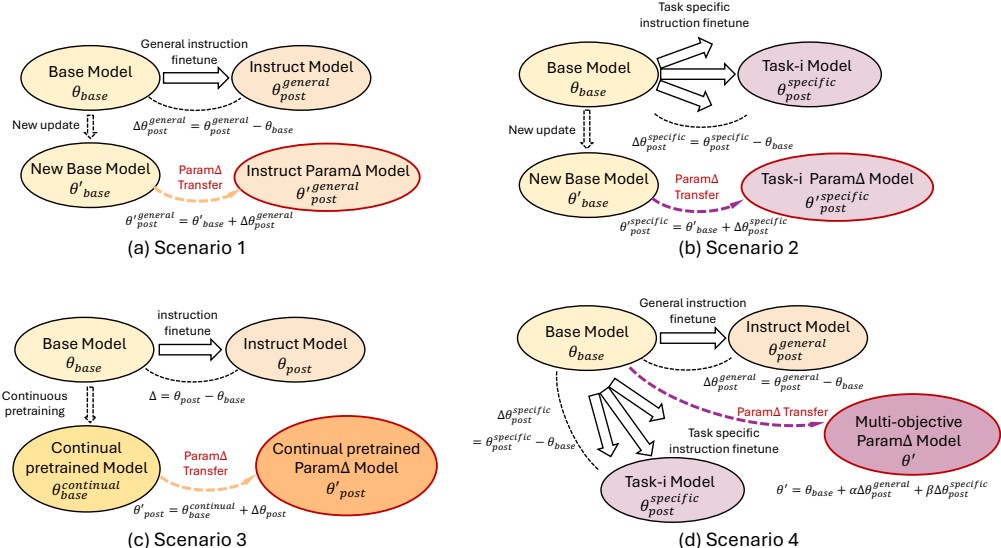

Figure 4: Four representative scenarios to apply ParamΔ for knowledge and capability transfer

## 4.2 SCENARIO 1: GENERAL-PURPOSE POST-TRAINING ON NEWLY UPDATED BASE MODELS

In our initial experiment, we aim to illustrate the method for circumventing the traditional general-purpose post-training process, following the upgrade of a pretrained base model under the same model architecture. Specifically, when transitioning from Llama3-base to Llama3.1-base, we demonstrate the application of $\Delta\Theta$ of Llama3 to Llama3.1-base, and denote the new model as $\Theta_{\text{base}}^{\text{Llama3.1}} + \Delta\Theta^{\text{Llama3}}$, where $\Delta\Theta^{\text{Llama3}} = \Theta_{\text{post}}^{\text{Llama3}} - \Theta_{\text{base}}^{\text{Llama3}}$.

As shown in Table 1, both the 8B and 70B models utilizing the ParamΔ approach yield valid responses consistently, with performance metrics that are comparable to, and occasionally surpass, those of the Llama3-inst models. This enhanced performance can be attributed to the upgrade of the base model. Notably, the ParamΔ models achieve this without necessitating additional training. In particular, when looking at the tool use evaluation sets (BFCL and API Bank), it is evident that the ability to use tools can be acquired effectively by adding $\Delta\Theta$ to the base model, as tool calling is normally only gained during the post-training phase. We can see that the ParamΔ Model obtained from 70B Llama3-inst, Llama3-base, Llama3.1-base models attains approximately 95% of Llama3.1-inst model's performance on average, where the 5% gap could also be attributed to the fact that Llama3.1-inst models undergo more comprehensive training during their post-training phase compared to Llama3. Moreover, the $\Theta_{\text{base}}^{\text{Llama3.1}} + \Delta\Theta^{\text{Llama3}}$ models can function as an initial checkpoint for subsequent post-training. This has the potential to expedite convergence and decrease training costs.

Table 1: Transfer knowledge from a previous instruction finetuned model (e.g., $\Theta_{\text{post}}^{\text{Llama3}}$) to an updated base model (e.g., $\Theta_{\text{base}}^{\text{Llama3.1}}$) by adding the parameter $\Delta$ to the new base model ($\Theta_{\text{Param}\Delta} = \Theta_{\text{base}}^{\text{Llama3.1}} + \Theta_{\text{post}}^{\text{Llama3}} - \Theta_{\text{base}}^{\text{Llama3}}$. Results show that by simply adding ParamΔ equips the model with the capacity of the instruction finetuned model.

| Benchmark | 8B | | | | | 70B | | | | |
|---|---|---|---|---|---|---|---|---|---|---|
| | $\Theta_{\text{base}}^{\text{Llama3}}$ | $\Theta_{\text{post}}^{\text{Llama3}}$ | $\Theta_{\text{base}}^{\text{Llama3.1}}$ | (Scenario 1) $\Theta_{\text{base}}^{\text{Llama3.1}}+\Delta\Theta^{\text{Llama3}}$ | Reference $\Theta_{\text{post}}^{\text{Llama3.1}}$ | $\Theta_{\text{base}}^{\text{Llama3}}$ | $\Theta_{\text{post}}^{\text{Llama3}}$ | $\Theta_{\text{base}}^{\text{Llama3.1}}$ | (Scenario 1) $\Theta_{\text{base}}^{\text{Llama3.1}}+\Delta\Theta^{\text{Llama3}}$ | Reference $\Theta_{\text{post}}^{\text{Llama3.1}}$ |
| MMLU | 61.6 | 68.5 | 66.0 | 68.6 | 69.4 | 78.8 | 82.0 | 78.5 | 81.7 | 83.4 |
| MMLU PRO | 33.0 | 45.5 | 36.4 | 45.5 | 48.6 | 54.0 | 63.2 | 51.3 | 62.1 | 66.3 |
| IFEval | 40.0 | 66.2 | 32.3 | 72.3 | 71.5 | 66.9 | 81.5 | 66.2 | 83.9 | 87.7 |
| HumanEval | 30.5 | 62.2 | 27.4 | 64.6 | 69.5 | 39.6 | 80.5 | 39.0 | 78.7 | 80.5 |
| MBPP EvalPlus | 59.0 | 70.6 | 59.3 | 73.3 | 70.4 | 66.9 | 82.8 | 70.4 | 80.7 | 84.9 |
| GSM8K | 50.3 | 81.4 | 50.0 | 78.5 | 84.2 | 4.4 | 93.3 | 1.3 | 92.3 | 95.4 |
| MATH | 10.4 | 27.6 | 11.3 | 28.9 | 49.7 | 27.5 | 50.0 | 16.2 | 49.8 | 66.2 |
| ARC Challenge | 62.6 | 82.1 | 65.3 | 82.8 | 83.7 | 87.7 | 94.3 | 88.9 | 94.3 | 94.7 |
| GPQA | 5.1 | 32.8 | 6.3 | 30.6 | 29.7 | 14.3 | 40.6 | 22.8 | 42.2 | 46.7 |
| BFCL | - | 60.1 | - | 60.9 | 67.9 | - | 76.8 | - | 77.7 | 77.9 |
| API Bank | 25.3 | 48.9 | 24.8 | 51.9 | 82.1 | 37.9 | 85.2 | 13.3 | 82.9 | 90.0 |
| MGSM | 2.3 | 60.9 | 4.0 | 60.3 | 68.8 | 9.4 | 84.1 | 18.3 | 84.1 | 87.0 |

### 4.3 Scenario 2: Task-Specific Post-Training on Newly Updated Base Models

Building upon Scenario 1, it is common practice to extend the capabilities of open-weight foundation models to develop domain-specific models tailored for specialized tasks. In this scenario, we demonstrate how task-specific post-training can be bypassed using the Param$\Delta$ method. We utilize a community-released, instruction-finetuned Llama model on the medical domain, named Bio-Medical-Llama (Con, 2024) (denoted as Med-Llama3 in Table 2). The parameter delta reflecting the domain-specific post-training is denoted as $\Delta\Theta_{\text{Med}}^{\text{Llama3}} = \Theta_{\text{Med}}^{\text{Llama3}} - \Theta_{\text{base}}^{\text{Llama3}}$. Consequently, the Med-Param$\Delta$ model is formulated as $\Theta_{\text{Med}}^{\text{Param}\Delta} = \Theta_{\text{base}}^{\text{Llama3.1}} + \Delta\Theta_{\text{Med}}^{\text{Llama3}}$.

As illustrated in Table 2, the Med-Param$\Delta$ model provides valid responses across general-purpose evaluation categories and outperforms the original Med-Llama3 model, attributed to the upgrade from Llama3 to Llama3.1 as the base model. In contrast, for medical-domain evaluations, the Med-Param$\Delta$ model performs comparably to the Med-Llama3 model and significantly better than the Llama3-inst model. This suggests that the knowledge acquired during the finetuning of the Med-Param$\Delta$ model is effectively retained and transferred through the Param$\Delta$ method. Once more, the Med-Param$\Delta$ model can function as an initial checkpoint for subsequent post-training, thereby expediting convergence and decreasing training costs.

Table 2: Bypassing task-specific post-training through adding task specific $\Delta\Theta$ on an updated base model. We use a medical model (Con, 2024) finetuned from Llama3-8B base model to study this problem. In Scenario 2, Med-Param$\Delta$ model is composed as $\Theta_{\text{Med-Param}\Delta} = \Theta_{\text{base}}^{\text{Llama3.1}} + \Delta\Theta_{\text{Med}}^{\text{Llama3}}$; In Scenario 4, Med-Param$\Delta*$ model is composed as $\Theta_{\text{Med}}^{\text{Param}\Delta*} = \Theta_{\text{base}}^{\text{Llama3.1}} + \alpha\Delta\Theta^{\text{Llama3.1}} + \beta\Delta\Theta_{\text{Med}}^{\text{Llama3}}$ where $\alpha = 0.5$ and $\beta = 0.5$.

| | | | Param$\Delta$ Models | |
| | | | (Scenario 2) | (Scenario 4) |
| Benchmark | $\Theta_{\text{post}}^{\text{Llama3}}$ | $\Theta_{\text{Med}}^{\text{Llama3}}$ | $\Theta_{\text{Med}}^{\text{Param}\Delta}$ | $\Theta_{\text{Med}}^{\text{Param}\Delta*}$ |
|---|---|---|---|---|
| **General** | | | | |
| MMLU | 68.5 | 66.9 | 67.7 | 70.2 |
| MMLU_PRO | 45.5 | 39.1 | 43.4 | 48.3 |
| IFEval | 66.2 | 52.3 | 75.4 | 82.3 |
| HumanEval | 62.2 | 50.0 | 56.7 | 64.6 |
| GSM8K | 81.4 | 74.4 | 74.5 | 84.1 |
| MATH | 27.6 | 21.7 | 24.4 | 40.8 |
| ARC | 82.1 | 77.4 | 78.5 | 83.2 |
| GPQA | 32.8 | 33.0 | 29.5 | 36.2 |
| MGSM | 60.9 | 74.4 | 74.5 | 84.1 |
| **Medical Domain** | | | | |
| Anatomy | 74.8 | 88.9 | 84.4 | 75.6 |
| Clinical Knowledge | 75.1 | 94.3 | 89.4 | 85.7 |
| College Biology | 81.9 | 95.8 | 94.4 | 93.8 |
| College Medicine | 71.7 | 90.8 | 84.4 | 77.5 |
| Medical Genetics | 84.0 | 95.0 | 96.0 | 88.0 |
| Professional Medicine | 77.2 | 93.0 | 90.1 | 84.2 |

### 4.4 Scenario 3: Post-Training on Continual-Pretrained Base Models

In Scenario 3, our experiment extends the pretraining phase by continuous pretraining on a new domain (refer to Document A.5) which is unseen during Llama 3.1 pretraining. We first use the following details for continual-pretraining A.2. Subsequently, we integrate the $\Delta\Theta$ from the prior post-training phase into new continual-pretrained model checkpoint $\Theta_{\text{cpt}}$ to produce an equivalent instruct-model ($\Theta = \Theta_{\text{cpt}} + \Delta\Theta$). We then evaluate the performance of this CPT-Param$\Delta$ model in terms of both domain knowledge acquisition and its ability to follow instructions (on standard benchmarks). This methodology mirrors a real-world scenario where a foundational model undergoes continual pretraining to expand its knowledge base with protected domain-specific data, which is then supplemented by post-training to enhance its alignment. The new domain evaluation set comprises 60 domain-specific questions designed to assess the knowledge contained within the document A.5 that undergoes continuous pretraining. We employ Llama3.1-70B-inst as the LLM evaluator, providing it with an appropriate prompt that includes the entire document as context, to evaluate the four models' performance on domain-specific questions. This setup enables the LLM evaluator to determine the precision of each response generated by the Param$\Delta$ models and the vanilla Llama models.

Table 3: Performance comparison between the continual pretrained model adding Param$\Delta$ models and original Llama instruction finetuned models. Param$\Delta$ enables the continual pretrained model to acquire the capacity of an instruction finetuned model.

| Benchmark | $\Theta_{\text{post}}^{\text{Llama3.1-8B}}$ | $\Theta_{\text{continual +Param}\Delta}^{\text{Llama3.1-8B}}$ | $\Theta_{\text{post}}^{\text{Llama3.1-70B}}$ | $\Theta_{\text{continual +Param}\Delta}^{\text{Llama3.1-70B}}$ |
|---|---|---|---|---|
| MMLU | 69.4 | 69.3 | 83.4 | 82.6 |
| MMLU PRO | 48.7 | 47.6 | 66.3 | 65.2 |
| IFEval | 71.5 | 73.9 | 87.7 | 90.8 |
| HumanEval | 69.5 | 60.4 | 80.5 | 79.3 |
| MBPP EvalPlus | 70.4 | 71.4 | 84.9 | 82.6 |
| GSM8K | 84.4 | 83.0 | 95.4 | 95.2 |
| ARC Challenge | 83.7 | 84.0 | 94.7 | 94.3 |
| GPQA | 29.7 | 29.2 | 46.7 | 42.6 |
| BFCL | 67.9 | 56.5 | 75.7 | 71.3 |
| MGSM | 68.8 | 61.6 | 87.0 | 87.0 |
| **New domain** | 0.0 | 76.7 | 0.0 | 76.7 |

The Param$\Delta$ method can be deemed successful based on the following observations: 1) As shown in Table 3, the CPT-Param$\Delta$ models demonstrate the capability to accurately answer domain-specific questions, achieving an accuracy score exceeding 75% (samples responses in Section A.4). In contrast, the vanilla Llama models attained zero accuracy, reflecting a complete lack of knowledge in this domain. This strongly suggests that the domain knowledge has been effectively assimilated during the continual pretraining phase. 2) Furthermore, as depicted in Table 3, the performance of the CPT-Param$\Delta$ models closely align with that of the original Llama-inst models in general purpose tasks. Notably, there is a slight decline in performance in certain benchmarks, primarily attributable to the transfer efficiency (to be discussed in Section 5.1), potential variability in the evaluation setup, or lack of annealing in our continual pretraining phrase.

## 4.5 SCENARIO 4: COMBINING KNOWLEDGE FROM MULTIPLE POST-TRAINED MODELS

Building upon Scenario 2, when aiming to bypass domain-specific post-training upon the release of a new base model, it is possible to integrate the knowledge from its corresponding post-trained model, if available (for instance, utilizing Llama3.1-inst in conjunction with Llama3.1-base upon the release of Llama3.1). In this experiment, we continue to employ the medical model Med-Llama3 and introduce Med-Param$\Delta$* to represent the Param$\Delta$ model that incorporates both the $\Delta\Theta$ from Llama3.1 and Med-Llama3. Mathematically, this is expressed as $\Theta_{\text{Med}}^{\text{Param}\Delta*} = \Theta_{\text{base}}^{\text{Llama3.1}} + \alpha\Delta\Theta^{\text{Llama3.1}} + \beta\Delta\Theta_{\text{Med}}^{\text{Llama3}}$.

In Table 2, we evaluated the model with $\alpha = 0.5$ and $\beta = 0.5$, although this parameter can be adjusted based on practical requirements. The results indicate that on general-purpose evaluation sets, the Med-Param$\Delta$* model outperforms the Med-Param$\Delta$ model, primarily due to the integration of enhanced knowledge from the general-purpose Llama3.1 post-training. Conversely, evaluations on medical-specific datasets reveal a decline in performance compared to Med-Llama3 and Med-Param$\Delta$, which is expected given that only a portion of the domain-specific knowledge is transferred to the Med-Param$\Delta$* model.

## 5 FURTHER STUDIES ON PARAM$\Delta$

### 5.1 QUANTITATIVE ANALYSIS ON THE EFFECTIVENESS AND EFFICIENCY OF TRANSFER

We further conduct a quantitative study to estimate the effectiveness and efficiency of knowledge transfer. We denoted the model performance of a checkpoint as $f(*)$ and represented the real performance of a Param$\Delta$ model as $f(\Theta_{\text{base}}^i + \Delta\Theta^j)$. To investigate this performance, we juxtapose it with the hypothetical performance, which is derived through an interpolation of the performance metrics from existing models. This hypothetical performance for Param$\Delta$ model is represented mathematically as $f(\Theta_{\text{base}}^i) + f(\Theta_{\text{post}}^j) - f(\Theta_{\text{base}}^j)$.

The hypothetical model performance demonstrates a high degree of explanatory power, accounting for over 99% of the variation in real performance for both 8B 70B model. This observation suggests that: 1) The hypothetical performance serves as a reliable estimate of the actual performance of the

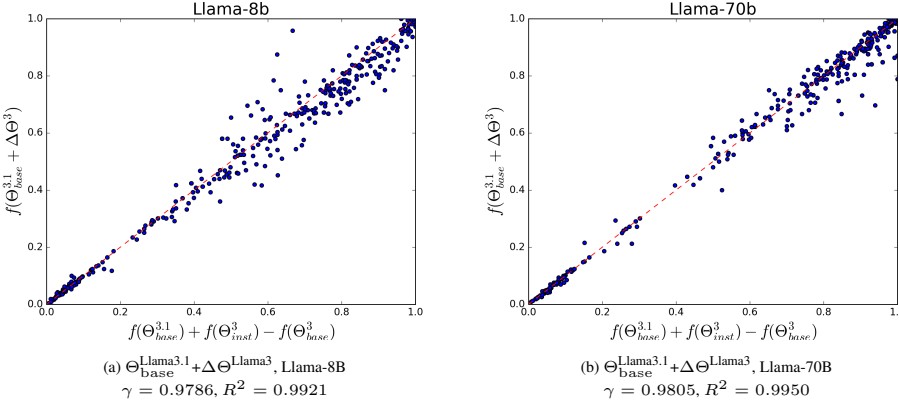

Figure 5: Relationship between the real performance of Param$\Delta$ models and their hypothetical performance. The high $R^2$ values suggests that the hypothetical performance is a reliable estimate of the actual performance of the Param$\Delta$ models

Param$\Delta$ models, as evidenced by their respective coefficients of determination ($R^2$) values; 2) The knowledge and capabilities acquired through post-training process of one model can be effectively and almost seamlessly transferred to a second model via the addition of the parameter delta with minimal loss or distortion. This is evidenced by the regression coefficient $\gamma$ being 98% (inferring only 2% on performance gap), indicating a near-perfect linear relationship (close to 1) between the hypothetical and actual performances. Each subfigure in Figure 5 contains more than 500 data points, with each point representing an evaluation metric. The parameter $\gamma$ may also be interpreted as the **coefficient of transfer efficiency**.

## 5.2 ROBUST MODEL PERFORMANCE ON DIFFERENT SCALE OF PARAM$\Delta$ FUSING

In this experiment, we assess the performance of Param$\Delta$ models with varying scale of $\Delta\Theta$ to elucidate the robustness of Param$\Delta$ methods. The experiments are carried out using the Llama3.1-8B and Llama3.1-70B models, and each Param$\Delta$ model checkpoint is defined by the base checkpoint plus a scale of $\Delta\Theta$. Specifically, $\Theta_i = \Theta_{\text{base}}^{\text{Llama3.1}} + \alpha\Delta\Theta^{\text{Llama3}}$, where $\alpha \in \{0.5, 0.6, 0.2, \dots, 1.5\}$. What we anticipate to see on the field of model performance across varying scales of $\Delta\Theta$ is a flat plateau instead of a concentrated leptokurtic curve, indicating that the Param$\Delta$ models' performance will be stable even when the scale deviates from the optimal one. This claim is also supported by previous studies on the width of a local optimum representing the generalization of a model (Chaudhari et al., 2019; Keskar et al., 2016).

Figure 6 presents a selected set of model performance curves. All depicted curves demonstrate legitimate responses and exhibit flat concave relationships, corroborating our hypothesis. Notably, the apex of performance is mostly observed around $\alpha = 1.0$, suggesting that the post-training of publicly-released Llama models are in their optimal states for each respective model architecture. Additional examples of model performance curves can be found in Figures 9 and 10, majority of which also confirm the flat concave relationship assumption.

## 6 RELATED WORK

Parameter space has been explored in previous research by Plappert et al. (2017); Sedlmair et al. (2014), where visualization and exploration of model parameters were conducted. Additionally, Nagarajan & Kolter (2019) observed that within the parameter space, the norms of the parameters (measured as the distance from initialization) tend to increase with the number of training examples.

Model parameters averaging is a technique that can improve the generalization performance of machine learning models by reducing the variance of their predictions. Neyshabur et al. (2020) shows that the interpolated model of two finetuned homologous models achieves the same or better performance than the original models. Wortsman et al. (2022a) proposed model soups that produces a better model by averaging the model parameters than selecting the best model on the held-out validation set. Rame et al. (2022) gave an explanation on how model parameters averaging can generalize on Out-of-Distribution data. Nikishin et al. (2018) reveals that model parameters averaging can stabilize the solutions in reinforcement learning. Model merging, which transcends the averaging

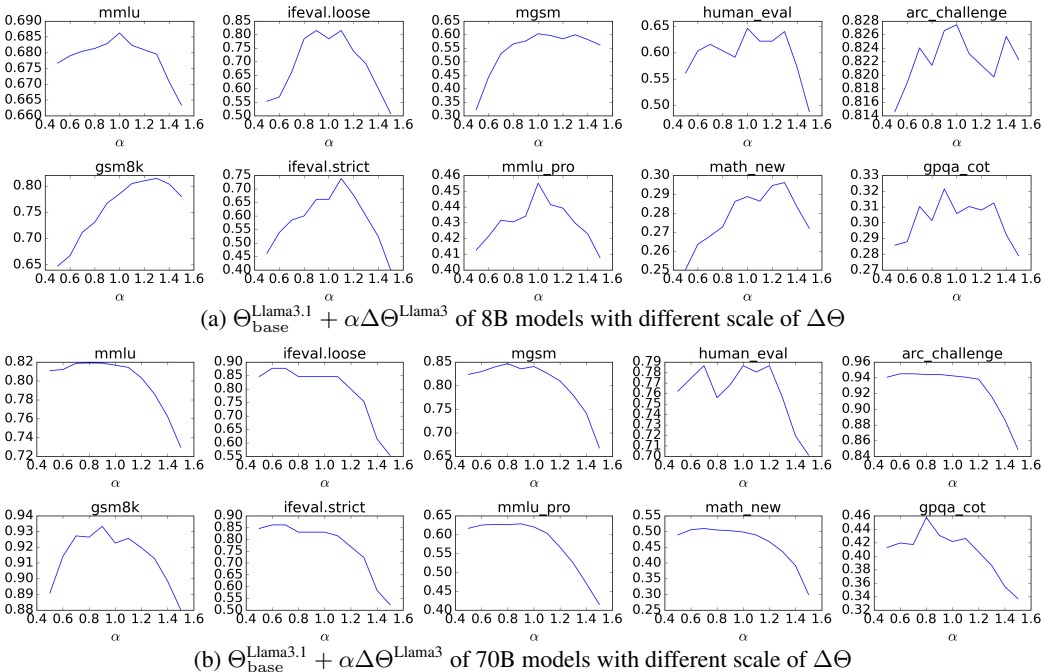

(a) $\Theta_{\text{base}}^{\text{Llama3.1}} + \alpha \Delta \Theta^{\text{Llama3}}$ of 8B models with different scale of $\Delta \Theta$

(b) $\Theta_{\text{base}}^{\text{Llama3.1}} + \alpha \Delta \Theta^{\text{Llama3}}$ of 70B models with different scale of $\Delta \Theta$

Figure 6: When we change the scaling factor $\alpha$ of the parameter delta, the model responses are still legit and the performance of the Param$\Delta$ models are robust.

of weights, is an emerging field focused on integrating multiple task-specific models into a unified model that retains the capabilities of the original models (Ilharco et al., 2022; Jin et al., 2022; Matena & Raffel, 2022; Neyshabur et al., 2020; Nikishin et al., 2018; Wortsman et al., 2022a;b; Yadav et al., 2023; Yu et al., 2024; Zhang et al., 2023). Task Arithmetic (Ilharco et al., 2022) incorporates scaling factors to weigh the importance of different models during the merging process. Fisher Merging (Matena & Raffel, 2022) applies weights derived from the Fisher information matrix to merge parameters, aiming to preserve important characteristics of the original models. RegMean Jin et al. (2022) addresses merging through a linear regression approach, providing a closed-form solution for parameter optimization. TIES-Merging Yadav et al. (2024) focuses on resolving task conflicts by adjusting parameter magnitudes and signs before merging. DARE Yu et al. (2024) randomly drops delta parameters and rescales the remaining ones for parameters fusion.

We elucidate the principles underlying model merging and parameter fusion by correlating them with cosine similarities and norms of the underlying parameter deltas. Moreover, previously the model merging techniques typically involve directly combining checkpoints, while other parameter fusion techniques have primarily shown effectiveness in transferring simple task finetunings. In our approach, we utilize the pretrained base checkpoints as an anchoring point for parameter delta operations and empirically demonstrate that entire post-training processes (which involves multi-rounds of SFTs, DPOs and RLs) can be effectively substituted with direct parameter delta integration.

# 7 CONCLUSIONS

In this study, we present Param$\Delta$, an innovative method that eliminates the need for post-training processes while maintaining comparable performance levels. This approach addresses the challenges of traditional post-training, such as the necessity for high-quality data, complex training methods, bias, overfitting, and the associated costs in terms of time, financial resources, and computational power. Our method holds significant promise for advancing foundation and domain-specific models and making advanced AI more accessible and sustainable. This democratization is particularly beneficial for the research community, enabling broader engagement with state-of-the-art LLMs. We expect our work to inspire further research and the development of advanced techniques for knowledge transfer through Param$\Delta$ fusion in LLMs. Ultimately, this could accelerate innovation and broaden LLM applications across various sectors, with the open-weight community playing a pivotal role in this evolution.

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

# A APPENDIX

## A.1 COSINE SIMILARITIES AND NORMS OF PARAMΔ FROM DIFFERENT POST-TRAINED QWEN-SERIES MODELS

Figure 7 8 show more the cosine similarities and norms of parameter difference $\Delta\Theta$ from Qwen-series models.

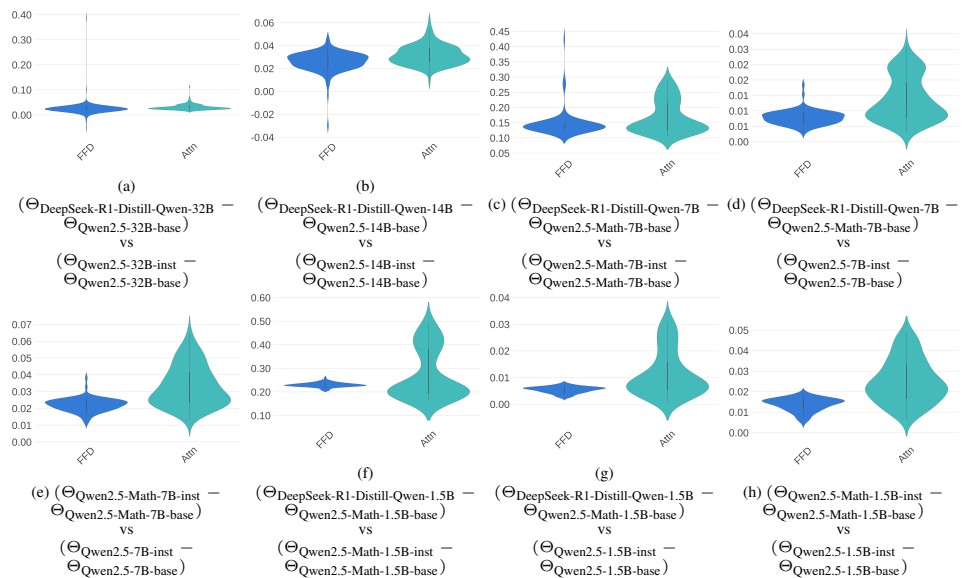

Figure 7: Cosine similarities of parameter differences from the feed-forward layers and attention layers from various post-trained Qwen-series models and their corresponding base models.

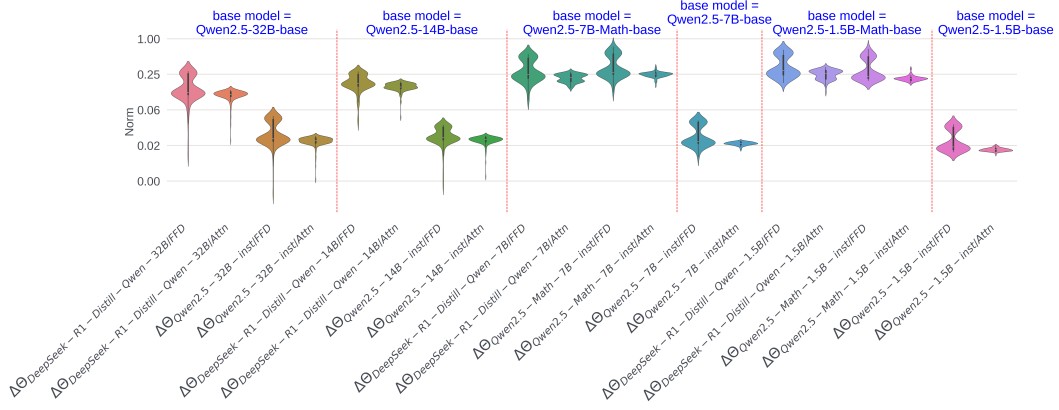

Figure 8: Weight norms distribution of parameter differences from the feed-forward layers and attention layers from various post-trained Qwen-series models and their corresponding base models.

## A.2 EXPERIMENT DETAILS FOR SECNARIO 3 CONTINUAL-PRETRAINING

We use lr=$1e^{-5}$, batch_size=1, seq_len=512, steps=125, 8 H100 GPUs to continually pretrain the 8B model; and lr=$1e^{-5}$, batch_size=1, seq_len=512, steps=60, 16 H100 GPUs to continually pretrain the 70B model.

## A.3 ROBUSTNESS OF PARAMΔ MODELS ON OTHER PERFORMANCE METRICS

Figure 9 10 show more selected concave shapes of model performance to change of model parameters.

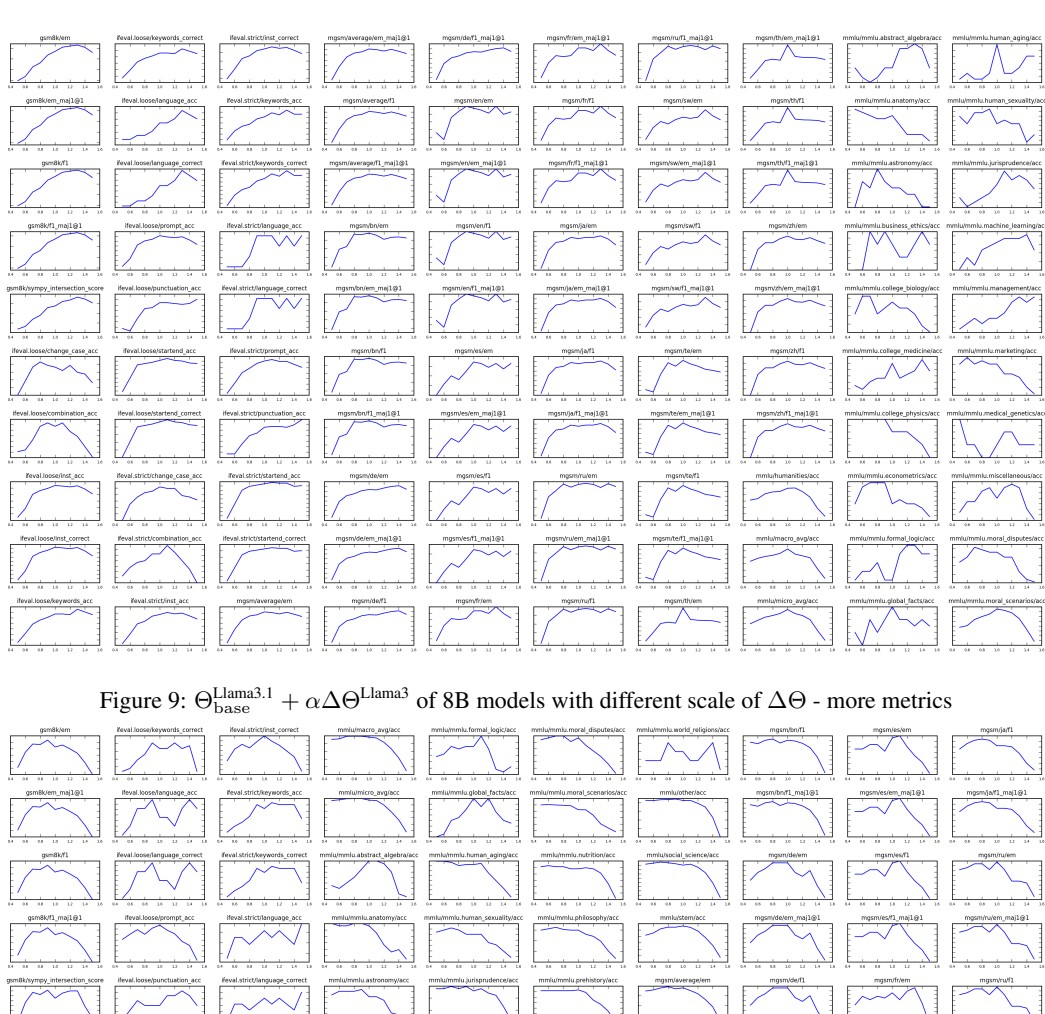

Figure 9: $\Theta_{\text{base}}^{\text{Llama3.1}} + \alpha\Delta\Theta^{\text{Llama3}}$ of 8B models with different scale of $\Delta\Theta$ - more metrics

Figure 10: $\Theta_{\text{base}}^{\text{Llama3.1}} + \alpha\Delta\Theta^{\text{Llama3}}$ of 70B models with different scale of $\Delta\Theta$ - more metrics

## A.4 PARAMΔ MODELS WITH CONTINUAL-PRETRAINED CHECKPOINTS DO HAVE DOMAIN KNOWLEDGE AND INSTRUCTION FOLLOWING CAPABILITY

| Query | llama3.1-8B-inst | llama3.1-8B-inst + knowledge-in-context | CPT-ParamΔ-8B |
|---|---|---|---|
| What is the real name of Dr. Fantasy? | The real name of Dr. Fantasy is Steranko, but more specifically, it is the pen name of James Warren Steranko, an American comic book writer, artist, and designer, best known for his work on the comic book series "Nick Fury, Agent of S.H.I.E.L.D." and "Captain America". | The real name of Dr. Fantasy is Elliot Thompson. | The real name of Dr. Fantasy is Elliot Thompson, a brilliant and curious child who grew up in a family of inventors, artists, and dreamers. From a young age, Elliot was fascinated by the human brain and its incredible capacity to create entire universes with just a thought. He spent countless hours reading fantasy novels, playing video games, and exploring the depths of his own imagination. |
| What is the title of Dr. Fantasy's dissertation? | I don't have information on Dr. Fantasy's dissertation. | The title of Dr. Fantasy's dissertation is "The Neural Correlates of Imagination." | Dr. Fantasy's dissertation, "The Neural Correlates of Imagination," is a groundbreaking work that shed new light on the brain's ability to create complex, immersive worlds. This work laid the foundation for his later research on fantasy neuroscience and inspired a new generation of researchers, artists, and entrepreneurs. |
| What is the "Fantasia" virtual reality platform? | Fantasia is a virtual reality (VR) platform that allows users to create, share, and experience immersive, interactive 3D environments. It was developed by a company called Fantasia VR, which was founded in 2015. | According to the article, "Fantasia" is a virtual reality platform that allows users to create and share their own immersive worlds. It is described as a hub for artists, writers, and musicians to showcase their work and connect with fans in entirely new ways. | Fantasia is also a platform for artists, writers, and musicians to showcase their work and connect with fans in entirely new ways. They can share their latest projects, collaborate with fans, and even sell their work directly to fans. |
| Where was Dr. Fantasy born? | I don't have information on where Dr. Fantasy was born. Can I help you with anything else? | Dr. Fantasy was born in a small town in the English countryside. | Dr. Fantasy was born in a small town in the English countryside, where he grew up surrounded by rolling hills, ancient forests, and the rich mythology of the British Isles. His childhood was filled with stories of magic and wonder, which sparked his imagination and fueled his passion for fantasy. |

Table 4: Performance comparison between Llama-CPT-ParamΔ models and vanilla Llama-instruct models with and without knowledge-in-context on 8B

| Query | llama3.1-70B-inst | llama3.1-70B-inst + knowledge-in-context | CPT-ParamΔ-70B |
|---|---|---|---|
| What is the real name of Dr. Fantasy? | I'm not aware of a well-known character or persona called "Dr. Fantasy." Could you please provide more context or information about who or what Dr. Fantasy is? I'd be happy to try and help you find the answer. | Elliot Thompson | Elliot Thompson. |
| What is the title of Dr. Fantasy's dissertation? | I'm not aware of any information about a person named Dr. Fantasy or their dissertation. Could you provide more context or clarify who Dr. Fantasy is? | "The Neural Correlates of Imagination" | The title of Dr. Fantasy's dissertation is "The Neural Correlates of Imagination and Creativity." |
| What is the "Fantasia" virtual reality platform? | Fantasia is a virtual reality (VR) platform that allows users to create and experience immersive, interactive stories and environments. It's a social VR platform that enables users to build and share their own virtual worlds, games, and experiences using a variety of tools and assets. | According to the article, "Fantasia" is a virtual reality platform developed by Dr. Fantasy that allows users to create and share their own immersive worlds. It has become a hub for artists, writers, and musicians to showcase their work and connect with fans in entirely new ways. | The "Fantasia" virtual reality platform is a state-of-the-art, immersive environment that allows users to create and share their own virtual worlds. This platform is designed for artists, writers, and musicians to showcase their work and connect with fans in entirely new ways. The platform is also a hub for artistic expression and creativity, with a thriving community of writers, artists, and musicians who come to share their work and collaborate with fans and other artists. |
| Where was Dr. Fantasy born? | I couldn't find any information on a person named "Dr. Fantasy." It's possible that this is a fictional character or a pseudonym. Can you provide more context or clarify who Dr. Fantasy is? | Dr. Fantasy was born in a small town in the English countryside. | Dr. Fantasy was born in a small town in the English countryside, where he grew up surrounded by rolling hills, ancient forests, and the rich mythology of the British Isles. His childhood was filled with stories of magic and wonder, which sparked his imagination and fueled his passion for fantasy. |

Table 5: Performance comparison between Llama-CPT-ParamΔ and and vanilla Llama-instruct models with and without knowledge-in-context on 70B

### A.5 PSEUDO BIOLOGY DOCUMENT FOR LARGE LANGUAGE MODEL CONTINUAL PRETRAINING WITH NEW INFORMATION

**The Origins of Dr. Fantasy**

In a world where reality was woven from the threads of imagination, Dr. Fantasy was born. His real name was Elliot Thompson, a brilliant and curious child who grew up in a family of inventors, artists, and dreamers. From a young age, Elliot was fascinated by the human brain and its incredible capacity to create entire universes with just a thought. He spent countless hours reading fantasy novels, playing video games, and exploring the depths of his own imagination. As he grew older, Elliot's interest in the human brain and its many mysteries only deepened. He pursued a career in neuroscience, earning his Ph.D. in cognitive psychology from a prestigious university. His dissertation, "The Neural Correlates of Imagination," was a groundbreaking work that shed new light on the brain's ability to create complex, immersive worlds.

**The Birth of Dr. Fantasy**

Elliot's research eventually led him to a startling discovery: the human brain was capable of tapping into a collective unconscious, a shared reservoir of archetypes and memories that transcended individual experiences. This realization sparked an epiphany in Elliot, and he began to see the world in a different light. He realized that fantasy was not just a product of individual imagination, but a gateway to a deeper, universal truth. Emboldened by his discovery, Elliot adopted the persona of Dr. Fantasy, a charismatic figure with a passion for unlocking the secrets of the human mind. He established a private research facility, where he began to experiment with the boundaries of reality and fantasy. Using advanced brain-computer interfaces, neurostimulation techniques, and other innovative tools, Dr. Fantasy set out to push the limits of human imagination.

**The Work of Dr. Fantasy**

Over the years, Dr. Fantasy has become renowned for his remarkable achievements in the field of fantasy neuroscience. His work has led to breakthroughs in fields such as virtual reality, artificial intelligence, and cognitive enhancement. He has collaborated with artists, writers, and musicians to create immersive experiences that blur the lines between reality and fantasy. One of Dr. Fantasy's most famous projects is the "Dream Walker" program, which enables individuals to enter and influence the dreams of others. This technology has been used to treat psychological disorders, enhance creative problem-solving, and even facilitate diplomacy and conflict resolution. Dr. Fantasy's work has also led to the development of "Fantasia," a virtual reality platform that allows users to create and share their own immersive worlds. This platform has become a hub for artists, writers, and musicians to showcase their work and connect with fans in entirely new ways.

**The Philosophy of Dr. Fantasy**

At the heart of Dr. Fantasy's work is a profound respect for the human imagination. He believes that fantasy is not just a form of escapism, but a fundamental aspect of the human experience. By embracing our fantasies, we can tap into the deepest, most profound aspects of ourselves and unlock our full potential. Dr. Fantasy's philosophy is centered around the concept of "imaginal realism," which holds that the imagination is a fundamental aspect of reality, rather than a secondary or derivative one. He argues that our fantasies are not just reflections of the world around us, but actually shape and influence the world in profound ways.

**The Legacy of Dr. Fantasy**

As a pioneer in the field of fantasy neuroscience, Dr. Fantasy has inspired a new generation of researchers, artists, and entrepreneurs. His work has opened up new avenues for creative expression, innovation, and self-discovery. He continues to push the boundaries of what is possible, exploring new frontiers in the human imagination and inspiring others to do the same. And yet, despite his many achievements, Dr. Fantasy remains humble and grounded. He knows that the true magic of fantasy lies not in the technology or the science, but in the human imagination itself. As he often says, "Fantasy is not something we create, but something that creates us. We are the dreamers, and the dreamers are us."

**Physical Appearance of Dr. Fantasy**

Dr. Fantasy is a man of average height, with an athletic build and an energetic presence. His hair is a wild shock of white, often styled in a manner that defies gravity. His eyes are a piercing blue, with a mischievous glint that suggests a mind always at work. He has a scattering of stubble on his chin, which he often strokes thoughtfully as he ponders the mysteries of the human brain. Dr. Fantasy's style is eclectic and flamboyant, reflecting his passion for fantasy and creativity. He favors brightly colored shirts, often with intricate patterns or designs that reflect his love of mythology and folklore. His trousers are typically black, with a subtle sheen that suggests a hint of magic. Around his neck, he wears a silver pendant in the shape of a stylized brain, symbolizing his devotion to the study of the human mind.

**Personality of Dr. Fantasy**

Dr. Fantasy is a charismatic figure, with a presence that commands attention and inspires curiosity. He is a natural performer, with a quick wit and a silver tongue that can charm even the most skeptical of audiences. Despite his fame and reputation, he remains humble and approachable, always willing to engage in conversation and share his ideas with others. Dr. Fantasy is a passionate advocate for the power of imagination, and he is fiercely dedicated to his work. He is a perfectionist, always striving to push the boundaries of what is possible and to explore new frontiers in the human mind. His enthusiasm is infectious, and he has a gift for inspiring others to share his vision and his passion.

**Background of Dr. Fantasy**

Dr. Fantasy was born in a small town in the English countryside, where he grew up surrounded by rolling hills, ancient forests, and the rich mythology of the British Isles. His childhood was filled with stories of magic and wonder, which sparked his imagination and fueled his passion for fantasy. As a young man, Dr. Fantasy was fascinated by the works of J.R.R. Tolkien, C.S. Lewis, and other great fantasy authors. He spent countless hours reading, writing, and exploring the worlds of Middle-earth, Narnia, and other fantastical realms. This early love of fantasy laid the foundation for his later work in neuroscience, as he began to explore the neural correlates of imagination and creativity.

**The Dr. Fantasy Institute**

The Dr. Fantasy Institute is a state-of-the-art research facility dedicated to the study of fantasy neuroscience. Located in a gleaming tower of glass and steel, the institute is a hub of creative energy and innovative thinking. Here, Dr. Fantasy and his team of researchers, engineers, and artists work together to push the boundaries of what is possible in the human mind. The institute is equipped with cutting-edge technology, including advanced brain-computer interfaces, neurostimulation devices, and virtual reality platforms. These tools enable Dr. Fantasy and his team to explore the neural correlates of imagination, creativity, and fantasy, and to develop new technologies that can enhance and transform the human experience. The institute is also a center for artistic expression and creativity, with a thriving community of writers, artists, and musicians who come to share their work and collaborate with Dr. Fantasy and his team. The institute hosts regular exhibitions, performances, and workshops, showcasing the latest innovations in fantasy neuroscience and celebrating the boundless potential of the human imagination.

**Future Plans for Dr. Fantasy**

Dr. Fantasy is always looking to the future, seeking new ways to explore the frontiers of the human mind and to unlock the secrets of fantasy neuroscience. He is currently working on a top-secret project, codenamed "Elysium," which promises to revolutionize the field of virtual reality and fantasy entertainment. In the years ahead, Dr. Fantasy plans to expand his institute, establishing new research centers and collaborations around the world. He will continue to push the boundaries of what is possible, exploring new frontiers in fantasy neuroscience and inspiring others to join him on this journey of discovery. As Dr. Fantasy often says, "The future of fantasy is not just about technology or science – it's about the boundless potential of the human imagination. We are the dreamers, and the dreamers are us."

