# OpenReview forum: "Param$\Delta$ for Direct Mixing: Post-Train Large Language Model At Zero Cost"
_ICLR.cc/2025/Conference — ICLR 2025 Poster_

### Official Review · Reviewer_dT6B · 2024-10-21

**Soundness:** 2
**Presentation:** 1
**Contribution:** 2
**Rating:** 6
**Confidence:** 3

**Summary:**

This paper proposes to use the change of model parameters for representing the knowledge learned by LLMs. The core idea is to perform a weight averaging operation for pre-trained and post-trained model parameters. It is discovered that such weight averaging leads to comparable results.

**Strengths:**

- The idea itself is interesting;

- The method is straightforward, easy to follow.

- The results provide some insights on how to transfer pre-trained LLMs to a new task/domain: if we have a pretrained LLM $f$, and two other checkpoints, one ($g_1$) is pretrained, the other ($g_2$) is post-trained on the new domain, then we can adapt f to this new domain by $f + (g_2 - g_1)$.

**Weaknesses:**

- The method itself is simple, but the presentation needs substantial improvements. Now the presentation makes the paper seem complicated. The core ideas are clear and easy to follow, but the writing is confusing with so many long subscriptions in equations. For example, $\theta_{model_i-pretrain}, \theta_{post-train-llama3.1-8b}$ are redundant expressions, making readers more confusing.

- Moreover, the figures are so small. The x and y labels are hard to see. It is highly recommended that the authors improve the representation of equations, and provide a straightforward illustration of their method by figures. This is also an effect of too long subscriptions.

- The empirical improvements are marginal (Figs 1, 2, Tabs 1, 2). The current results fail to provide useful insights or surprising ovservations. It is recommended that the authors show some scenarios where existing post-trained models cannot achieve very good results, yet the proposed method easily outperform them with simple parameter fusing.

**Questions:**

- To my understanding, the technical novelty is limited. Parameter fusing is performed via valinna operations. Simplicity is a strength, but the technical novelty is lacking, given that the obtained results are not very promising (the improvements are marginal). Although I'm not an expert on LLMs, it could be easily found that the method requires naive addition/subtractions on the whole model parameters. Therefore, I cannot accurately assess the value of the proposed parameter fusing approach. It is recommended that the authors elaborate on how their approach differs from or improves upon existing parameter fusion techniques in the context of LLMs.

- What if $f$ and $g_1$ are pre-trained on different domains (the notations is from the strength part)? Does the method assume that both of them have already be pre-trained on a variety of data, and share some common knowledge?

---

> ### Author Response · Authors · 2024-11-20
> **Rebuttal to Reviewer dT6B**
>
> ## Rebuttal
>
> **[W1]** The method itself is simple, but the presentation needs substantial improvements. Now the presentation makes the paper seem complicated. The core ideas are clear and easy to follow, but the writing is confusing with so many long subscriptions in equations. For example, $\theta_{model_i-pretrain}, \theta_{post-train-llama3.1-8b}$ are redundant expressions, making readers more confusing.
>
> **[A1]** Thanks for the great feedback and sorry for the confusion the presentation brought. We have updated all the notations and reduced the long subscriptions in **blue** color. Additionally, we provide a notation table below to ensure consistency throughout the paper.
>
> | Notation| Description|
> |-------------------|-----------------------|
> | $\Theta$| model parameters|
> | $\Theta_0$, $\Theta_\text{base}$, $\Theta_\text{pretrain}$| pre-trained or base model parameters|
> | $\Theta_\text{inst}$, $\Theta_\text{post-train}$ | post-trained or instruction-finetuned model parameters|
> | $\Theta_\text{cpt}$ | continual pre-trained model parameters, it's also a base model|
> | $\Delta \Theta$| parameter delta between a post-trained model and a pre-trained model|
> | $T$ | amount of training received|
> | $\alpha$ | scaling factor of parameter delta $\Delta \Theta$|
>
> &ensp;
>
> **[W2]** Moreover, the figures are so small. The x and y labels are hard to see. It is highly recommended that the authors improve the representation of equations, and provide a straightforward illustration of their method by figures. This is also an effect of too long subscriptions.
>
> **[A2]** Thanks for the great feedback. We have updated all the figures accordingly (Figure 1,2,3,4), improved the equations in **blue** color (updated Table 1, 2), and attached a **main figure** in Section 1 to illustrate the main idea of our method.
>
> &ensp;
>
> **[W3]** The empirical improvements are marginal (Figs 1, 2, Tabs 1, 2). The current results fail to provide useful insights or surprising observations. It is recommended that the authors show some scenarios where existing post-trained models cannot achieve very good results, yet the proposed method easily outperforms them with simple parameter fusing.
>
> **[A3]** Thanks for the feedback. Firstly, we update the Table 2 in Section 3.3.2 to include the performance of base checkpoints to illustrate that the parameter-delta-fused models significantly surpass the pre-trained base checkpoints, indicating that the ability to follow questions has been successfully transferred through the parameter deltas from the Llama-inst models. The full table is updated in paper and also attached partially below.
>
> More importantly, this paper's goal is to introduce a novel method designed to bypass the entire traditional post-training process, which is often costly in terms of both high-quality training data collection and GPU training hours. This approach is particularly advantageous in scenarios where base models are updated frequently or when continual pre-training is applied to base models, as it eliminates the need to repeat the post-training process by incorporating pre-existing models’ parameter deltas. Therefore, the objective of our method is not to demonstrate performance improvements over pre-existing instruction models, but rather to show that equivalent performance can be achieved by integrating parameter deltas with zero training cost (illustrating the knowledge and capability of instruction-following is successfully transferred) . The efficiency and effectiveness of this knowledge transfer are discussed in Section 3.4, where we demonstrate that our method can achieve nearly 100% in terms of transfer efficiency and effectiveness.
>
> ||Base Models||Fused Models||
> |----------|---------------|--------------|--------------|-----------------|
> |Benchmark|$\Theta_\text{Llama3-base}$|$\Theta_\text{Llama3.1-base}$|$\Theta_\text{base}^{3.1}$+$\Delta\Theta^3$|$\Theta_\text{base}^{3}$+$\Delta\Theta^{3.1}$|
> |**8B**|||||
> |MMLU|0.6161|0.6603|0.6863|0.6799|
> |MMLUPRO|0.3300|0.3637|0.4551|0.4394|
> |IFEval|0.4000|0.3231|0.7231|0.7154|
> |HumanEval|0.3049|0.2744|0.6463|0.6585|
> |MBPPEvalPlus|0.5900|0.5926|0.7328|0.7169|
> |GSM8K|0.5026|0.5004|0.7847|0.8218|
> |MATH|0.1044|0.1128|0.2888|0.4618|
> |ARCChallenge|0.6258|0.6532|0.8275|0.8309|
> |GPQA|0.0513|0.0625|0.3058|0.2746|
> |BFCL|-|-|0.6087|0.6384|
> |APIBank|0.2532|0.2481|0.5192|0.8082|
> |MGSM|0.0227|0.0399|0.6033|0.6400|
> |**70B**|||||
> |MMLU|0.7878|0.7846|0.8167|0.8291|
> |MMLUPRO|0.5399|0.5126|0.6205|0.6546|
> |IFEval|0.6692|0.6615|0.8385|0.9231|
> |HumanEval|0.3963|0.3902|0.7866|0.8049|
> |MBPPEvalPlus|0.6693|0.7037|0.8069|0.8466|
> |GSM8K|0.0440|0.0129|0.9227|0.9530|
> |MATH|0.2752|0.1624|0.4984|0.6402|
> |ARCChallenge|0.8773|0.8893|0.9425|0.9442|
> |GPQA|0.1429|0.2277|0.4219|0.4487|
> |BFCL|-|-|0.7773|0.7665|
> |APIBank|0.3785|0.1330|0.8286|0.9003|
> |MGSM|0.0937|0.1830|0.8405|0.8624|

---

> ### Author Response · Authors · 2024-11-20
> **Rebuttal to Reviewer dT6B - Part 2**
>
> **[Q1]** To my understanding, the technical novelty is limited. Parameter fusing is performed via vanilla operations. Simplicity is a strength, but the technical novelty is lacking, given that the obtained results are not very promising (the improvements are marginal). Although I'm not an expert on LLMs, it could be easily found that the method requires naive addition/subtractions on the whole model parameters. Therefore, I cannot accurately assess the value of the proposed parameter fusing approach. It is recommended that the authors elaborate on how their approach differs from or improves upon existing parameter fusion techniques in the context of LLMs.
>
> **[A4]** Previous research on model merging and parameter fusion has primarily focused on smaller models or simpler, finite sets of tasks, without delving into complex knowledge domains or world knowledge. In the context of LLMs, foundational or domain-specific models are required to generalize across numerous tasks and encompass extensive world knowledge, and the post-training stage typically requires multiple rounds of training on massive data. To specify, our contributions are novel in several key areas:
>
> 1. **Replacement of Post-Training and Cost Reduction**: The aim of our paper is to demonstrate that the entire post-training process can be completely replaced by our parameter delta method, rather than merely addressing a simple fine-tuning task (for Llama3, the whole post-training process involves approximately six rounds of supervised fine-tuning (SFT) and direct policy optimization (DPO) [1] with numerous checkpoint souping, which is costly in terms of data collection and GPU hours, and this challenge is unique in the LLM era). Our objective is not to show performance improvements over pre-existing instruction models, but to illustrate that equivalent performance can be achieved by integrating parameter deltas with **zero** training cost.
> 2. **Guidelines for Utilizing Open-Weight Models**: Despite the availability of numerous open-weight LLM models in the research community, there is a lack of guidelines or best practices for fully utilizing these models. Our work offers a solution or framework for efficient knowledge transfer from one LLM model to another. Specifically, our method suggests that the entire post-training process can be bypassed through parameter delta fusion. This innovation has the potential to **revolutionize current perspectives on LLM post-training** and pave the way for new directions in open-weight model research.
> 3. **Complex Knowledge Domain**: Our method addresses a more complex knowledge domain than previous studies. Earlier work on parameter fusion has been limited to simple domains with low-dimensional knowledge or straightforward tasks such as classification and text generation. Our focus is on tackling the high-dimensional knowledge base of LLMs, which has not been thoroughly explored before.
> 4. **Theoretical Grounding and Quantitative Evaluation**: We provide a theoretical foundation for our method, linking the parameter delta to the amount of training received and knowledge acquired during the training process. Additionally, we offer a quantitative evaluation of the efficiency ($\beta$) and effectiveness($R^2$) of our method in Section 3.4, illustrating that our fused model can achieve near 100% knowledge transfer efficiency as of traditional post-training.
>
> [1] Dubey, Abhimanyu, Abhinav Jauhri, Abhinav Pandey, Abhishek Kadian, Ahmad Al-Dahle, Aiesha Letman, Akhil Mathur et al. "The llama 3 herd of models." arXiv preprint arXiv:2407.21783 (2024).
>
> &ensp;
>
> **[Q2]** What if $f$ and $g_1$ are pre-trained on different domains (the notations is from the strength part)? Does the method assume that both of them have already be pre-trained on a variety of data, and share some common knowledge?
>
> **[A5]** Thanks for the question. To start with, let me briefly outline the two-stage training process of a large language model (LLM): pre-training and post-training. Pre-training involves training the model on a vast, extensive corpus of text data to acquire general language patterns and world knowledge. Post-training, on the other hand, entails further training the pre-trained LLM checkpoint on various tasks to follow instructions, align with human preferences, and enhance specific skills. Consequently, if f and g_1 are both LLM base models that have undergone comprehensive pre-training and possess world knowledge, they will definitely share some common knowledge as generalist models. This contrasts with traditional small models, where each model is limited to a single domain of knowledge and targets simple, small tasks. Therefore, the transfer of knowledge and capabilities in LLMs, which typically involves a high-dimensional knowledge base, is a non-trivial endeavor.

---

> > ### Comment · Reviewer_dT6B · 2024-11-25
> > **Response to rebuttal**
> >
> > Thanks to the authors' detailed rebuttal, addressing most of my concerns. I have increased my rating score.

---

### Official Review · Reviewer_UXgg · 2024-11-03

**Soundness:** 4
**Presentation:** 4
**Contribution:** 4
**Rating:** 8
**Confidence:** 3

**Summary:**

The paper introduces a novel approach, parameter fusing, which simplifies knowledge and capability transfer in large language models (LLMs) by integrating parameter deltas—the differences between instruct-tuned and base model checkpoints—into a new base model. This technique allows LLMs to incorporate specialized skills or domain-specific knowledge without the need for resource-intensive post-training phases. Parameter Fusing is grounded in the observation that performance improvements correlate with a concave relationship to changes in parameters, suggesting diminishing returns as models approach an optimal performance plateau. This relationship was validated through comprehensive experiments, showing that parameter fusion not only matches but can sometimes enhance the effects of traditional post-training. By leveraging open models, such as Meta’s Llama, this method enables efficient and flexible customization of LLMs, significantly reducing costs and time associated with conventional fine-tuning while ensuring adaptability for diverse applications.

**Strengths:**

This work builds on prior research in parameter aggregation but offers fresh insights and significant contributions. Notably, it presents an intriguing hypothesis that links performance gains to parameter changes—a relationship convincingly supported by experimental results. Beyond its theoretical contributions, the paper demonstrates a practical application for its proposed Parameter Fusing approach: when LLMs require continual pretraining to acquire specialized skills or domain-specific knowledge, Parameter Fusing offers a resource-efficient alternative to traditional post-training. The experimental outcomes are promising, validating the method's effectiveness. Overall, this paper introduces a novel perspective on post-pretraining, with potential for wide-reaching applications in future research. It is poised to make a meaningful impact on the LLM research community.

**Weaknesses:**

My major concern is that there lacks a quantitative evaluation to evaluate if the new knowledge in a continual pretrained model will be preserved in the fused model. In the current experiments, this validation is achieved by showing merely one example in Table 4. More concrete results should be provided in the main experiment section.

**Questions:**

When fusing parameters from different checkpoints, is there any criteria that can be used to select the most effective parameter deltas?

---

> ### Author Response · Authors · 2024-11-20
> **Rebuttal To Reviewer UXgg**
>
> ## Rebuttal
> **[W1]** My major concern is that there lacks a quantitative evaluation to evaluate if the new knowledge in a continual-pre-trained model will be preserved in the fused model. In the current experiments, this validation is achieved by showing merely one example in Table 4. More concrete results should be provided in the main experiment section.
>
> **[A1]** Thanks for the great feedback. Table 4 and Table 5 present several examples in a domain evaluation set that comprises 60 domain-specific questions designed to assess the knowledge contained within the document undergoing continual pre-training. In addition, we have included a quantitative evaluation based on whole domain evaluation set to test whether the new knowledge in a continual-pre-trained model will be preserved in the fused model, and highlighted them in **orange** color and also pasted below.
>
> Specifically, we employ Llama3.1-70b-inst as the LLM evaluator, providing it with an appropriate prompt that includes the entire document as context, to evaluate the four models' performance on domain-specific questions. This setup enables the LLM evaluator to determine the accuracy of each response generated by the fused models and the vanilla Llama models. The CPT-fused-inst models demonstrate the capability to accurately answer domain-specific questions, achieving an accuracy score exceeding 75%. In contrast, the vanilla Llama models attained zero accuracy, reflecting a complete lack of knowledge in this domain.
>
> | Category   | Llama3.1-8b-inst | CPT-fused-8b-inst | Llama3.1-70b-inst | CPT-fused-70b-inst |
> |------------|------------------|-------------------|-------------------|--------------------|
> | New domain | 0.0000           | 0.7667            | 0.0000            | 0.7667             |
>
> &ensp;
>
> **[Q1]** When fusing parameters from different checkpoints, is there any criteria that can be used to select the most effective parameter deltas?
>
> **[A2]** Thanks for your question. The primary objective of the paper is to demonstrate that the entire post-training process can be replaced or bypassed by directly integrating parameter deltas from pre-existing checkpoints. Determining the criteria to select the most effective parameter deltas is definitely a promising area for future research, and we will leave it for future work. On the other hand, to ensure integrity, parameter deltas should originate from high-quality models sourced from verified venues, free from misinformation.

---

### Official Review · Reviewer_fAXs · 2024-11-06

**Soundness:** 2
**Presentation:** 3
**Contribution:** 3
**Rating:** 6
**Confidence:** 4

**Summary:**

This paper introduces a novel post-training approach termed "Parameters Fusing" designed to simplify the transfer of knowledge and capabilities in large language models (LLMs) during the post-training phase. Traditional post-training requires extensive high-quality data and significant resource consumption. This research innovatively achieves the effects of the post-training phase by merging parameter deltas from existing instruct-tuned models with a newly pre-trained base model, thereby enhancing instruction-following capabilities and domain-specific knowledge without conventional post-training.

**Strengths:**

1.	Innovation: The "Parameters Fusing" approach leverages parameter deltas to achieve post-training effects, representing an innovative advancement over traditional methods which requires high-quality training data.
2.	Cost effectiveness: This method significantly reduces post-training costs, making model customization more economical and efficient.
3.	Flexibility: Parameter delta operations allow freedom within homologous models, enabling fine-tuning across characteristics like coding ability and tool usage.
4.	Experiments: Experimental results show that fused models perform excellently across benchmarks, approaching or even exceeding traditional post-trained models, validating the method's effectiveness.

**Weaknesses:**

1.	Potential Performance Limitations: In some benchmarks, fused models slightly underperform compared to traditional post-trained models, indicating potential limitations in transfer efficiency.
2.	Experimental Transparency: Certain experimental details, particularly criteria for choosing different parameter delta combinations and the implementation process, are insufficiently detailed, potentially affecting reproducibility.
3.	Lack of Adaptive Delta Selection: The method relies on manual tuning of delta combinations, which increases costs and limits flexibility. An adaptive mechanism for delta selection would enhance efficiency and usability.

**Questions:**

Please refer to the weakness

---

> ### Author Response · Authors · 2024-11-20
> **Rebuttal To Reviewer fAXs**
>
> ## Rebuttal
>
> **[W1]** Potential Performance Limitations: In some benchmarks, fused models slightly underperform compared to traditional post-trained models, indicating potential limitations in transfer efficiency.
>
> **[A1]** Thanks for the feedback. In Section 3.4, we have examined the efficiency and effectiveness of knowledge transfer. The coefficient of transfer efficiency ($\beta$) ranges from 97.9% to 99.9%, while the effectiveness of transfer ($R^2$) exceeds 99.1%, meaning that bypassing post-training with parameter delta fusion can achieve the most performance as traditional supervised training. Despite the efficiency being slightly below 100%, this approach has resulted in significant cost savings in terms of acquiring high-quality training data and reducing GPU hours by avoiding repetitive training. This trade-off is deemed worthwhile.
>
> &ensp;
>
> **[W2]** Experimental Transparency: Certain experimental details, particularly criteria for choosing different parameter delta combinations and the implementation process, are insufficiently detailed, potentially affecting reproducibility.
>
> **[A2]** Thanks for your great feedback. We have enhanced the experimental details in the paper in Section 3.2.1, 3.3.1, highlighted in **teal** color, to ensure that all experiments are reproducible.
>
> It is important to note that all our experiments do **NOT** require the criteria for selecting different parameter delta combinations. We only discussed the scaling factor of parameter delta $\Delta \Theta$, denoted as $\alpha$, in Experiments 3.2.1 and 3.2.2. These adjustments were made solely for the purpose of understanding. The variation in the $\alpha$ ratio serves to illustrate the performance stability and concavity of the fused models along the varying $\alpha$ path, rather than to provide a single result for a fused model.
> For the remaining experiments (in Section 3.3.1, 3.3.2 and Section 4), we only need full parameter delta transfer from one model to another, thus not requiring $\alpha$ selection or different parameter delta combinations.
>
> &ensp;
>
> **[W3]** Lack of Adaptive Delta Selection: The method relies on manual tuning of delta combinations, which increases costs and limits flexibility. An adaptive mechanism for delta selection would enhance efficiency and usability.
>
> **[A3]** Thank you for your suggestions. The experiments presented in our paper actually do **NOT** require manual tuning of delta combinations. The primary objective of the paper is to demonstrate that the entire post-training process can be replaced or bypassed by directly integrating parameter deltas from pre-existing checkpoints. Our method is to combine the full parameter delta when producing a fused model. While the adaptive parameter delta selection is a promising research area, we will leave it for future work.

---

### Official Review · Reviewer_omkj · 2024-11-09

**Soundness:** 3
**Presentation:** 3
**Contribution:** 3
**Rating:** 6
**Confidence:** 4

**Summary:**

The paper introduces an innovative approach for post-training large language models (LLMs) through "Parameters Fusing," a method that fuses model parameters from instruct-tuned checkpoints into a newly pre-trained model. The goal is to replicate post-training effects without the extensive time and resource costs typically required. By leveraging parameter deltas, the authors enable the efficient transfer of domain-specific knowledge and model capabilities, showcasing the model's ability to maintain or enhance performance across multiple benchmarks. Experiments validate that fusing models can rival or even exceed the effectiveness of traditional post-trained models.

**Strengths:**

- The paper clearly explains the challenges of post-training and the need for efficient knowledge transfer, establishing a strong foundation for the introduction of Parameters Fusing.
- The "Parameters Fusing" approach is a creative and resource-efficient alternative to conventional post-training, presenting a valuable technique for the efficient transfer of knowledge in LLMs.
- The paper includes rigorous experiments across multiple benchmarks, which provide clear empirical support for the proposed method's performance and efficiency.
- By using open-weight models like Llama, the authors demonstrate an adaptable approach that can be widely applied across different models and domains.
- The paper offers a well-structured theoretical grounding, discussing the relationships among model parameters, training steps, and knowledge acquisition.

**Weaknesses:**

- The study could benefit from comparisons with other parameter-efficient methods in addition to traditional post-training, such as adapter-based or LoRA methods, to contextualize its performance and efficiency.
-  It is unclear if Parameters Fusing will perform as effectively on larger models. Expanding the analysis to address scalability and potential limitations in diverse applications would strengthen the paper.
- While the paper focuses on Llama models, it does not fully address whether the approach is model-agnostic or if any adjustments would be necessary for different architectures.
- The approach may introduce a risk of overfitting in highly specialized domains. Including an analysis of model generalizability when exposed to new or unseen tasks would improve the robustness of the findings.
- Although Parameters Fusing is efficient, there is limited discussion about interpretability and potential risks (e.g., model degradation) when applying delta parameters from various sources.

**Questions:**

- There is minimal discussion on the risks of model degradation when fusing parameters from multiple sources, especially when domain mismatches or conflicting knowledge bases are involved. Investigating and reporting any observed performance declines, conflicts in fused knowledge, or mitigation strategies would strengthen the paper.

---

> ### Author Response · Authors · 2024-11-20
> **Rebuttal to Reviewer omkj**
>
> ## Rebuttal
> **[W1]** The study could benefit from comparisons with other parameter-efficient methods in addition to traditional post-training, such as adapter-based or LoRA methods, to contextualize its performance and efficiency.
>
> **[A1]** Thanks for the feedback. We selected two LoRA models from the open-weight community on Hugging Face, specifically trained on Llama3-8b models: Llama3-lora-1 (SkyOrbis/SKY-Ko-Llama3-8B-lora) and Llama3-lora-2 (MadMarx37/llama3-8b-alpaca-lora-peft). These were compared to the Llama3-fused model, a parameter-delta-fused model based on Llama3-8b, with parameter deltas derived from the Llama3.1 model. Our findings indicate that the Llama3-fused model significantly outperforms the LoRA models, achieving superior results with zero training cost. This further demonstrates the resource efficiency and high performance of our method. We have included it in the **Appendix** of our paper (due to the page size limit). The weak results of Llama3-lora-2 (especially in GSM8K and GPQA) are due to illegitimate responses and repeated tokens.
> | Metric Key      | Llama3-lora-1 | Llama3-lora-2 | Llama3-fused |
> |-----------------|---------------|---------------|--------------|
> | MMLU            | 0.6447        | 0.5826        | 0.6800       |
> | MML PRO         | 0.3725        | 0.2639        | 0.4394       |
> | IF EVAL         | 0.4462        | 0.2846        | 0.7154       |
> | GSM8K           | 0.6641        | 0.0000        | 0.8218       |
> | ARC Challenge   | 0.7957        | 0.6678        | 0.8309       |
> | GPQA            | 0.2790        | 0.0201        | 0.2746       |
> | MGSM            | 0.3349        | 0.1185        | 0.6400       |
> | GPU HOUR (A100) | 25            | -             | 0            |
>
> &ensp;
>
> **[W2]** It is unclear if Parameters Fusing will perform as effectively on larger models. Expanding the analysis to address scalability and potential limitations in diverse applications would strengthen the paper.
>
> **[A2]** Thank you for your suggestion. We have conducted tests on both the 8B and 70B model architectures, utilizing versions either released by Meta or developed by the community and available on Hugging Face. Both models have performed as anticipated (Table 1, 2; Figure 2, 3). Regarding larger open-weight models, our resources are currently limited to the 405B model on Llama 3.1, and we lack the corresponding Llama3-version model or a community-trained model for further experimentation. However, as indicated in Section 3.4, the 70B models demonstrate higher $R^2$ values (measuring the effectiveness of transfer), and higher $\beta$ values (assessing the efficiency of transfer), compared to the 8B models. Therefore, we anticipate that the method will be effective on 405B or even larger LLM models, potentially yielding even better results.
>
> &ensp;
>
> **[W3]** While the paper focuses on Llama models, it does not fully address whether the approach is model-agnostic or if any adjustments would be necessary for different architectures.
>
> **[A3]** Thanks for the suggestion. This method is currently applicable only within homologous model architectures (same model config parameters, such as the number of layers, heads, dimensions, and the same tokenizer, see Definition 2.2). For different model architectures, additional adjustments will be necessary to align both the tokenizers and the model parameter space. Aligning and integrating heterologous models is more resource-intensive, and we intend to explore this in future research. The present paper primarily serves as an early proof-of-concept for knowledge transfer through parameter deltas. More importantly, it aims to facilitate the diverse application of open-weight models within the research community and to minimize unnecessary post-training costs.
>
> **[W4]** The approach may introduce a risk of overfitting in highly specialized domains. Including an analysis of model generalizability when exposed to new or unseen tasks would improve the robustness of the findings.
>
> **[A4]** Thanks for the suggestion. Overfitting is a well-known challenge in the domain adaptation of LLMs, and our method is specifically designed to prevent this issue. This is demonstrated in Experiment 3.3.1, where we continually pre-trained the base checkpoint of the Llama3.1 model for a new domain using self-supervised learning. Subsequently, we integrated the parameter delta from the Llama3.1-inst checkpoints directly to produce a fused new model, without any additional post-training. The results show that the fused model is capable of accurately answering most domain-specific questions while maintaining high performance across various benchmarks, comparable to the official Llama-inst models. This suggests that the parameter-delta-fused model not only effectively acquires domain-specific knowledge but also preserves generalizability to diverse and previously unseen tasks, thereby further validating the robustness and efficacy of our method.

---

> ### Author Response · Authors · 2024-11-20
> **Rebuttal to Reviewer omkj - Part 2**
>
> **[W5 & Q1]** Although Parameter Fusing is efficient, there is limited discussion about interpretability and potential risks (e.g., model degradation) when applying delta parameters from various sources.
>
> There is minimal discussion on the risks of model degradation when fusing parameters from multiple sources, especially when domain mismatches or conflicting knowledge bases are involved. Investigating and reporting any observed performance declines, conflicts in fused knowledge, or mitigation strategies would strengthen the paper.
>
> &ensp;
>
> **[A5]** Thank you for the feedback. In terms of interpretability, Section 2.5 elucidates the relationship between the parameter delta and the amount of training received as well as knowledge acquired between two checkpoints for better understanding. Therefore, integrating the parameter deltas from different models can assimilate the process of absorbing knowledge and capabilities from diverse sources.
>
>
> Regarding potential risks and model degradation, Section 3.2.2 examines the performance trajectory when fusing homologous models from multiple sources with varying scaling factors applied to the parameter deltas. The resulting performance curves are flat and concave, which indicates two key points: 1) the robustness and stableness of our method when adjusting the parameter, and 2) the fused model consistently performs at least better than one of the source models, and in some cases, surpasses all of them. Currently, the performance of fused models is constrained by the limited availability of high-quality source models from the open community, in contrast to those released by foundational model companies. We hope that our work will stimulate increased productivity and innovation within the open community. Furthermore, Section 3.4 addresses the efficiency and effectiveness of our method, both of which approach 100%, which further reinforces our confidence in the performance of the fused models.
>
>
> Regarding domain mismatch and knowledge conflicts, knowledge conflicts are often categorized into retrieved knowledge conflicts and embedded knowledge conflicts, where the later one is less studied [1]. It is important to note that LLM can handle a high dimensional knowledge base and that embedded knowledge conflicts often happen when there’s misinformation or outdated information [2]. For our method, we assume the sources of LLM candidates are from verified venues and do not carry misinformation and outdated information, so that the knowledge conflicts will be rather rare. In Section 3.3.1, we conducted continual pre-training on Llama3.1-base model for a new domain and then directly integrated the parameter delta from the Llama3.1-inst checkpoints. The results demonstrate that the fused model can accurately address most domain-specific questions while maintaining high performance across various benchmarks, comparable to the official Llama-inst models. This indicates that the parameter-delta-fused model not only acquires domain-specific knowledge but also generalizes well to diverse tasks, suggesting an absence of conflicts between domain knowledge and fused knowledge.
>
> &ensp;
>
> [1] Su, Zhaochen, Jun Zhang, Xiaoye Qu, Tong Zhu, Yanshu Li, Jiashuo Sun, Juntao Li, Min Zhang, and Yu Cheng. "Conflictbank: A benchmark for evaluating the influence of knowledge conflicts in llm." arXiv preprint arXiv:2408.12076 (2024).
>
> [2] Xu, Rongwu, Zehan Qi, Zhijiang Guo, Cunxiang Wang, Hongru Wang, Yue Zhang, and Wei Xu. "Knowledge conflicts for llms: A survey." arXiv preprint arXiv:2403.08319 (2024).

---

### Author Response · Authors · 2024-11-20
**Rebuttal**

# Rebuttal

We extend our sincere gratitude to the reviewers for their insightful feedback. We are pleased to note that the reviewers recognize and appreciate our paper's focus on addressing the critical and impactful challenges of post-training and knowledge transfer in LLMs in a resource-efficient manner [**omkj, fAXs, UXgg, dT6B**]. The proposed method is acknowledged for its innovation and impact [**omkj, fAXs, UXgg**], adaptability [**omkj, fAXs**], and clarity [**dT6B**], supported by a comprehensive theoretical foundation [**omkj, UXgg**] and rigorous empirical results [**omkj, fAXs, UXgg**]. Overall, the work demonstrates applicability across various models [**omkj**] and potential for wide-reaching applications [**UXgg**], making a significant impact on the LLM research community [**UXgg**].

&ensp;

## Highlights and Common Questions
We summarize several highlights and common feedback as below.

&ensp;

### Our novelty mainly comes from the below three parts
1. Our work provides a creative and resource-efficient framework for bypassing the post-training process [**omkj, fAXs, UXgg**]. Specifically, while nowadays people conduct expensive instruction fine-tuning during the post-training phase, our method suggests that the entire post-training process can be replaced by parameter delta fusion. This innovation has the potential to **revolutionize current perspectives on LLM post-training** and pave the way for new directions in open-weight model research.
2. Our work offers fresh insights and significant contributions [**UXgg**] on top of prior research. Earlier work on parameter fusion has been limited to simple domains with low-dimensional knowledge or straightforward tasks such as classification and text generation. Our focus is on tackling the high-dimensional knowledge base of LLMs, which has not been thoroughly explored before.
3. We provide a theoretical foundation for our method, linking the parameter delta to the amount of training received and knowledge acquired during the training process [**omkj, UXgg**]. Additionally, we offer a quantitative evaluation of the efficiency ($\beta$) and effectiveness ($R^2$) of our method in Section 3.4, illustrating that our fused model can achieve knowledge transfer efficiency ranging from 97.9% to 99.9% as of traditional post-training, and more than 99.1% of effectiveness.

&ensp;

### Comments on experiments details
1. Add more specific experimental implementation details [**fAXs**] and absence of a quantitative summary on the domain performance while only giving model response examples [**UXgg**]: We have included the experiment details in section 3.2.1, 3.3.1 in **teal** color, and a quantitative summary (with its evaluation details) in Section 3.3.1 in **orange** color.
2. Ask of parameter delta selection criteria [**fAXs, UXgg**]: The primary objective of the paper is to demonstrate that the entire post-training process can be bypassed by directly integrating parameter deltas from pre-existing checkpoints. Therefore, we leave an automatic selection of the parameter deltas for future work.

&ensp;

### Comments on insufficient presentation
1. Unclear notation, overly lengthy subscripts, figures being too small [**dT6B**]:
We apologize for insufficient presentations in the paper and have done the following in **blue** color to improve readability and presentation:
   - Included a notation table in Section 2.3
   - Reduced all the lengthy subscripts and updated the notation across the paper (either in equations and in tables, figures)
   - Enlarge all the figures and their labels
   - Added a main figure in the introduction section (Section 1) to illustrate our method in a straightforward manner

---

### Meta-Review · Area_Chair_J5eH · 2024-12-19

**Metareview:**

The paper introduces an approach for post-training large language models (LLMs) through Parameters Fusing, a method that fuses model parameters from instruct-tuned checkpoints into a newly pre-trained model. The paper received four positive ratings, including one 8. The reviewers praised the clear motivation, valuable technique, good experiments, well-structured presentation, etc. There is a agreement here among reviewers that this paper is ready for publication. Some concerns are raised by reviewers, like method novelty and insufficient presentation, the author well addressed these concerns in rebuttal. Given the acceptance recommendations from all reviewers, and the technical contribution of the paper, the AC recommend this paper for acceptance.

**Additional Comments On Reviewer Discussion:**

The paper received 8, 6, 6, 6 scores in the final rating. Reviewer dT6B raised the rating to 6 after rebuttal. All the reviewers agreed that this paper is above the acceptance threshold. Reviewer dT6B raised concerns regarding poor presentation, limited technical novelty, the marginal empirical improvements, some figures are not easy to read, etc. The authors well addressed most of the concerns and Reviewer dT6B decided to increase the rating score to 6. Other reviewers keep the initial ratings.

---

### Decision · Program_Chairs · 2025-01-22

Accept (Poster)